# Can Nutritional Status in Adults Be Influenced by Health Locus of Control?

**DOI:** 10.3390/ijerph192315513

**Published:** 2022-11-23

**Authors:** Wojciech Gruszka, Aleksander J. Owczarek, Mateusz Glinianowicz, Monika Bąk-Sosnowska, Jerzy Chudek, Magdalena Olszanecka-Glinianowicz

**Affiliations:** 1Health Promotion and Obesity Management Unit, Department of Pathophysiology, Medical Faculty in Katowice, Medical University of Silesia in Katowice, 40-752 Katowice, Poland; 2Pathophysiology Unit, Department of Pathophysiology, Medical Faculty in Katowice, Medical University of Silesia in Katowice, 40-752 Katowice, Poland; 3Department of Psychology, Social Sciences and Humanities, School of Health Sciences in Katowice, Medical University of Silesia in Katowice, 40-752 Katowice, Poland; 4WSB Academy, Department of Health Sciences, Cieplaka 1C Str., 41-300 Dąbrowa Górnicza, Poland; 5Department of Internal Medicine and Oncological Chemotherapy, Medical Faculty in Katowice, Medical University of Silesia, 40-055 Katowice, Poland

**Keywords:** nutritional status, body weight, visceral obesity, health locus of control

## Abstract

The external health locus of control (HLC) is based on an assumption that obtained health results depend on the influences of external factors. As for the internal HLC—that is the effect of our own actions and capabilities. Little is known regarding how the HLC can influence body weight or the occurrence of visceral obesity. The study aimed to assess the relationship between the health locus of control and nutritional status in adults. The study included 744 adults (452 women, 292 men; 2.8% underweight, 43.8% normal weight, 29.7% overweight, and 23.7% obese). In addition to anthropometric measurement and socio-demographic data, the health locus of control, using the Multidimensional Health Locus of Control Scale (MHLC) by Wallston K, Wallston B, and DeVellis R, was assessed. The percentage of subjects with an internal HLC did not differ significantly between obese, overweight, and normal-weight groups. However, the percentage of subjects with an external HLC—dependent on the impact of others—was significantly higher in both men and women with obesity than in corresponding overweight and normal-weight groups (*p* < 0.01). Yet, the percentage of subjects with an external HLC subject to the impact of chance was significantly higher among overweight and obese women than in those of normal weight (*p* < 0.05) only. Women with overweight or obesity, with external health locus of control, experienced both the impact of others and of chance more often than women with normal weight. However, men with overweight and with obesity more often had external health locus of control influenced only by others.

## 1. Introduction

Obesity is recognized as one of the main causes of premature death in developed countries [1]. One of the consequences of obesity causing disability or premature death is ischemic and hemorrhagic stroke [2]. It has also been shown that obesity decreases the ability of older people to work [3]. The incidence of overweight and obesity, despite growing knowledge about their pathogenesis, continues to increase [4,5]. Psychological factors, including personality traits, may play an important role in the development of obesity by shaping eating behavior. The relationships between these factors and nutritional status are hardly known. Uncovering them may help to identify people who need proper psychotherapy to prevent weight gain. Furthermore, knowledge about these relationships among patients with overweight and obesity condition may increase the effectiveness of obesity treatment and long-term weight loss maintenance.

The probability of individuals’ behavior depends on their personality traits and learned behavior. One personality trait is the locus of control, which may be external or internal. This may be general or related to any area, such as the individual’s health. Based on the initial Rotter’s theory [6], internal and external subtypes of health locus of control (HLC) were described [7]. In 1973, Levenson [8] divided the external dimension of HLC into two subtypes according to dominant beliefs regarding the impact of other people and the impact of chance. The existence of an external HLC rests on the assumption that the obtained health result depends on external factors, for example, a physician, another person, or chance. In other words, health-related events are perceived as not completely dependent upon personal actions. In contrast, the presence of an internal HLC is based on the belief that health depends on the individual. Such a person believes in a causal relationship between actions taken and certain health-related events.

The internal HLC is considered to be more favorable because people who believe that they are responsible for their health are more often characterized by healthy behaviors. It has been observed that people with this type of HLC are more careful about oral hygiene, less often smoke cigarettes, more often self-examine breasts, and use contraception. They also seek health information, and if they get sick they have more knowledge about their disease than people with an external HLC and are more likely to follow diet and physical activity recommendations [9,10,11,12,13,14,15,16]. In addition, they can better appreciate social support [17,18]. In contrast, people with external HLC are more prone to experiencing stressful events [19] and have a higher level of depressive symptoms [20,21], more often denying having symptoms of a disease and diminishing the importance of recommendations for treatment or rehabilitation [19]. Furthermore, people with external HLC have a negative view of social support and poor self-assessment of health [22,23]. It has been observed that this type of HLC is more common in women, older adults and people of non-European origin. Also, people with external HLC more often have a lower level of education and socio-economic status [24].

All the above-described data suggest that internal HLC is associated with healthy behaviors and favors normal body weight, while external HLC promotes the development of overweight and obesity. However, to the best of our knowledge, data about relationships between nutritional status and health locus of control are still limited. One of the latest studies indicated that an external HLC with powerful other factors is positively associated with a body mass index (BMI) among normal-weight subjects. Furthermore, a study conducted online that assessed nutritional status based on self-reported weight and height showed that subjects with grades I and II obesity have higher levels of influence by powerful others than do subjects with overweight conditions [25]. The personality traits of health locus of control may be used to predict behaviors of patients with overweight and obesity and may be crucial in planning and improving the effectiveness of future interventions. So far, no studies have been conducted to assess the HLC in which categories of nutritional status are assessed on the basis of actual measurements of body weight and height in people of different ages. Therefore, the study aimed to assess the relationship between health locus of control and nutritional status in adults.

## 2. Materials and Methods

Eight-hundred-twenty-four respondents, aged 16 years or above, were enrolled in the study between June 2010 and August 2011. The respondents were recruited and invited to the study by co-authors who are physicians in their outpatient clinics. The reasons for visits were various, excluding reporting to the clinic for obesity treatment. Secondary obesity, mental illness (lifetime bipolar disorders, schizophrenia and current substance use disorder) and anorexia or bulimia were the exclusion criteria. The recruitment for the study is presented in Figure 1. 

Finally, a study group consists of 744 people, including 452 women (60.7%) and 292 men (39.3%). 

The respondents were not rewarded in any way for participating in the study. The basic characteristics of the study group are shown in Table 1. All participants in the study were informed about its objective and gave written consent to the study protocol. The study was approved by the Bioethics Committee of the Medical University of Silesia (KNW-0022/KB1/136/I/08). 

Body weight (without shoes, in light clothing, using the certified electronic RADWAG balance, with an accuracy of 0.1 kg) and height (in an upright standing position, without shoes, with an accuracy of 0.5 cm, using an integral part of RADWAG balance) were measured. BMI was calculated using the standard formula. Assessment of nutritional status was based on BMI according to World Health Organization (WHO) criteria [26]. Waist circumference was measured according to the International Diabetes Federation (IDF) guidelines [27]. The diagnosis of visceral obesity was based on IDF criteria [27].

The Multidimensional Health Locus of Control Scale (MHLC) by Wallston K, Wallston B and DeVellis R was used [28]. Translation to Polish language and validation of the scale was done by Juczyński [29]. MHLC was administered as a hard copy. The time to complete the questionnaire was not limited. The scores were calculated according to the recommendation [29]. MHLC consists of 18 items, scored from 1 to 6 pts., divided into three groups allowing assessment of internal and external HLC, and in external—the impact of chance and the impact of powerful others. No one score can be combined as a total score—it is theoretically possible to be high or low on all three scales [28,29]. The total score for each HLC dimension ranges from 6 pts. to 36 pts., with higher scores indicating an increasing level of health control. 

MHLC was utilized as the most widely used tool for assessing HLC in adults. It was translated into and validated in many languages, including Polish [29,30]. The Polish version of the MHLC was characterized by high psychometric values. The internal consistency of MHCL established on the basis of Cronbach’s alpha, was 0.74 for an internal HLC, 0.69 for an external HLC with the impact of chance, and 0.54 for an external HLC with the impact of others. Reliability (r) assessed by the test-retest method (after six weeks) for internal HLC was 0.72; for external HLC with an impact of chance was 0.60 and for external HLC with an impact of others was 0.64 [29]. 

### Statistical Analysis

Statistical analysis was performed with Statistica 13.0 software (Polish version).

The results were presented as: mean ± standard deviation for normally distributed data, median and upper and lower quartiles or range for data that deviate from the normal distribution, and percentages for data in nominal and ordinal scales. The assessment of distribution was based on the Shapiro-Wilk test. Comparison of variables in nominal and ordinal scales was performed using the χ^2^ test (when the Cochran condition was met) and χ^2^ for trend. To compare analyzed variables among groups with different nutritional statuses, concerning sex, a two-factor analysis of variance with the contrast analysis was used. The homogeneity of variance was assessed using the Levene test. In the case of non-compliance with the conditions of the parametric ANOVA test, a nonparametric equivalent—the ANOVA Kruskal-Wallis test was used. To assess the risk factors for overweight and obesity status, univariable and multivariable ordinal logistic regression was used. The multivariable model was the best one for all factors. The Brant test was used to assess the proportional odds assumption. The results were considered statistically significant with a *p*-value of less than 0.05.

## 3. Results

An underweight condition was diagnosed in 2.8% of participants, normal weight in 43.8%, overweight in 29.7%, and obesity in 23.7%. Visceral obesity was diagnosed in 290 (64.1%) women and 130 (44.5%) men. Table 1 shows the characteristics of the study group. Participants with overweight conditions (N = 221) and obesity (N = 176) were older than subjects with normal weight (N = 347), and participants with obesity were older than overweight ones.

There were statistically significant differences between groups in education level and marital status (*p* < 0.001) overall and in both men and women (*p* < 0.001). However, no differences between men and women in nutritional status groups were observed regarding education level, yet normal-weighted and overweight men lived more frequently alone than women (*p* < 0.01). Women and men with overweight conditions and obesity less often (*p* < 0.01) had higher education and lived alone less often (*p* < 0.001) than normal-weighted participants. No differences were observed between participants who were overweight and obese.

There were no statistically significant differences regarding alcohol consumption in women and men (*p* = 0.14 and 0.39, respectively) as well as in sports activities (*p* = 0.24 and *p* = 0.40, respectively). Moreover, in all nutritional status groups, alcohol consumption was more common among men than women (*p* < 0.001). 

There were significant differences in the frequency of smoking (*p* < 0.01), yet taking sex into account, this difference remained significant only for women (*p* < 0.05) but not for men (*p* = 0.18). Women with obesity smoked more frequently than the normal-weighted ones (*p* < 0.01). No other differences were observed.

A high internal HLC characterized 44.8% of respondents and 64.1% had a high external HLC (69.9% with a greater impact of others, 64.4% with a greater impact of chance). There were eight possible types of health locus of control: weak, undifferentiated—20.5% of subjects; strong, undifferentiated—12.8%; enlarging impact of chance—12.2%; diminishing impact of chance—10.2%; growing impact of others—12.5%; diminishing impact of others—7.1%; strong external—10.0%; strong internal—14.7%.

Women had a higher level of internal HLC (*p* < 0.05) and a lower level of external HLC with the impact of chance (*p* < 0.01) than men. There were no differences in the level of external HLC with the impact of others between women and men (*p* = 0.21).

### 3.1. Health Locus of Control and BMI

The score obtained by the subjects determining the internal HLC was similar in normal weight, overweight, and obese subgroups among both women and men—Figure 2.

We also checked the interactions between age and BMI. There were moderately significant interactions between age and BMI in both men and women (r = 0.35; *p* < 0.001 and r = 0.38; *p* < 0.001, respectively). 

Two-way analysis of variance showed a statistically significant association between body weight (*p* < 0.001) and an external HLC with the impact of others. The external HLC with the impact of others occurred significantly more often among overweight and obese participants than in the normal weight group, regardless of sex (*p* < 0.001). Considering sex, significantly higher scores for this type of HLC were found among women with overweight conditions and obesity than those with normal body weight (*p* < 0.05 and *p* < 0.001, respectively) and among women with obesity than overweight women (*p* < 0.01). Also, among the men, significantly higher scores for this type of HLC were found in participants with obesity than in overweight and normal-weight men (*p* < 0.01 and *p* < 0.001, respectively). However, there was no significant difference in this aspect between men who were overweight and those of normal weight (*p* = 0.2)—Figure 2.

Two-way analysis of variance showed a statistically significant association between sex (*p* < 0.01) but not with BMI (*p* = 0.15) and the external HLC with the impact of chance. The external HLC with the impact of chance occurred significantly more often in women with overweight or obesity than in men with overweight or obesity (*p* < 0.05). In addition, women with obesity obtained significantly higher scores than normal-weight women (*p* < 0.01)—Figure 2.

The percentage of subjects with internal HLC did not differ significantly between the obese, overweight, and normal-weight subgroups among both women (*p* = 0.64) and men (*p* = 0.62). The percentage of subjects with external HLC largely affected by the impact of others was significantly higher among participants with obesity than those with overweight and normal weight among both women (*p* < 0.01) and men (*p* < 0.01). The percentage of subjects with external HLC affected by the impact of chance was significantly higher among women with normal weight than among those with overweight or obesity (*p* < 0.05), but not among men (*p* = 0.94).

A significantly higher percentage of women with obesity was found among those with a strong external HLC and women with overweight among those with external HLC with the impact of chance. However, the percentage of women with normal weight was highest among women with an internal HLC and low external HLC with the impact of others (*p* < 0.05). In contrast, in men, there were no significant differences between the nutritional status of participants with different types of HLC (*p* = 0.31).

### 3.2. Health Locus of Control and Visceral Obesity

Two-way analysis of variance showed a statistically significant association between sex (*p* < 0.05), but for visceral obesity (*p* = 0.44) and the internal HLC, no significant interaction was noted—Figure 3.

Two-way analysis of variance showed a statistically significant association between visceral obesity (*p* < 0.001), but not between sex (*p* = 0.18) and the external HLC with the impact of others. Both women and men with visceral obesity had a higher level of external HLC with an impact of others than women and men without visceral obesity—Figure 3.

Two-way analysis of variance showed a statistically significant association between visceral obesity (*p* < 0.001) and not between sex (*p* < 0.01) and the external HLC with the impact of chance. Women with visceral obesity had a higher level of external HLC with the impact of chance than women without visceral obesity—Figure 3.

### 3.3. Univariable and Multivariable Ordinal Logistic Regression for Overweight and Obesity

Table 2 shows the results of the univariable and multivariable ordinal logistic regression for overweight and obesity status. In both women and men, the risk of being overweight and obese was higher in older subjects, frequent alcohol consumers, and those with high external HLC with the impact of others and of chance. In the multivariable ordinal logistic regression, only age and the external HLC with the impact of others proved to be statistically significant. Older age and the higher external HLC increased the risk of being overweight and obese.

## 4. Discussion

To the best of our knowledge, this is the first study that assessed not only a group of young subjects but also middle-aged and elderly ones. Moreover, the assessment of BMI was based on measurements, not on self-reported weight and height. Furthermore, it is the first study that analyzed the association between visceral obesity and HLC.

In the presented study we assessed the relationship between health locus of control and nutritional status. As a main result, it is clear that the external HLC with the impact of others may be a risk factor for the development of obesity. The second finding indicates that the external HLC with the impact of chance may favor the development of this disease only in women. These relations were confirmed in the analysis of visceral obesity diagnosed on the basis of measurement of waist circumference.

The obtained results are in line with previous observations which showed that persons with internal HLC more often represent healthy behaviors and are more prone to change their behaviors under the influence of education. As noted, these persons give themselves realistic goals and try to reach them. Furthermore, they show greater persistence in pursuing activities [9,10,11,12,13,14,15,16]. On the contrary, it seems that a high level of external HLC observed in subjects with overweight conditions or obesity, especially in women, may be associated with higher susceptibility to stress [19] and this may cause emotional eating [31]. Recent studies by Duplaga et al. [32] showed that the scores of secondary school adolescents with higher external HLC with the impact of others show that they are more likely to consume fast food, which supports our findings. An external HLC with the impact of others in participants with obesity seems to have a negative effect on changing lifestyle and lead to constant seeking for a “proper” physician. In contrast, Chavez et al. [33] showed that people with an internal HLC weighed more than those with an external HLC. Yet, in line with our results, Radcliff et al. [34] observed that adolescents aged 13–18 years with lower BMI had greater internal HLC. These differences are difficult to explain. Relationships between nutritional status and personality traits are not simple and there are probably other factors that can modify them.

Our results are also in line with those obtained by Cebolla et al. [25] that showed an association between the powerful others dimension of HLC and higher BMI. However, in this study levels of internal HLC and chance did not differ between participants who were overweight or obese. These partial differences may be explained, among others, by using other scales in the assessment and the younger ages of subjects in our study. This hypothesis is supported by our observations showing that the percentage of persons with obesity increases with age. Of note, we showed no previously reported sex differences and relationships between locus of control and visceral obesity, which is somehow distinct from obesity defined according to WHO based on BMI value [35]. The results of previous studies showed that the BMI value is less sensitive in the prediction of cardiovascular disease than waist circumference measurement, better reflecting visceral fat distribution [36]. In addition, in line with our results, Perdue et al. [37] found that women with obesity after bariatric surgery who still identify themselves as obese have more external HLC influenced by powerful other factors.

To our knowledge, most previous studies focused on relationships between internal HLC and healthy behaviors in patients with obesity. It has been shown that patients with obesity with internal HLC obtaining better results during obesity treatment were less likely to discontinue the treatment and more often maintain lower body weight [38,39,40]. However, not all studies confirm these observations [41,42].

The nature of the causal relationship between obesity and HLC remains unclear and needs further study. It seems that a bidirectional relationship is possible and both paths—“obesity is a cause of the external HLC” and “external HLC is a cause of obesity”—seem to be probable. The first path would primarily affect young people while the process of forming personality traits has not ended. In adults, this seems to be rather unlikely given that the locus of control is included in personality traits, and these are considered to be stable throughout the course of life. The probability of the path “obesity is the cause of the external HLC” can be supported by the hypothesis that HLC can change itself as a result of illness, especially one that has long-lasting consequences. It was observed that in patients with epilepsy or spinal cord injury the external HLC is stronger than in healthy individuals [43,44,45]. However, it cannot be ignored that these subjects primarily had an external HLC. Also, Neymotin and Nemzer [46] pointed out that the direction of causality from obesity to a more external locus of control is very likely due to discrimination by other people.

Presented results together with previous observations imply the need to consider the assessment of HLC when planning obesity treatment by physicians, psychologists, dieticians and other practitioners. A deeper understanding of the individual’s HLC seems to be useful in a person-oriented weight reduction approach [46]. Particularly, internally focused people should tend to succeed more often in obesity treatment and should be more effective in maintaining obtained results [15,47]. Externally focused subjects with high powerful other levels should benefit more from the assistance provided by the physician and other practitioners [37,48]. On the other hand, one might expect that the effects of obesity treatment depend only on the doctor. For patients with external HLC with a high chance level, a good healthcare provider may create a more internal HLC, leading to better adherence to recommendations [48]. However, this is still an area for future studies.

The presented study has numerous limitations. First, a three-dimensional model, used in our study, is the most common. However, there are models separating “influence other” factors in the HLC construct into two: physicians and proper “other people” [49], and they introduce the fourth dimension of God [50]. Second, the study enrolled subjects utilizing medical services, which limits the generalization of the obtained results for the general population and had a cross-sectional design without any follow-up. Third, the number of participants who were underweight in the study group was relatively small, so we decided to omit presenting results from this group to avoid misinterpretation. Fourth, to avoid repeating the categorization of nutritional status, we decided to use general WHO cut-off points for adults [26] rather than the reference values for juveniles [51], as the study included only three individuals between 16 and 18 years of age. Fifth, external factors such as culture and other psychological features (for example self-assessment or mood) that may interact with the observed associations [52] were not assessed. In addition, clinical data including the occurrence of obesity complications were not collected. Sixth, the presented data were collected in 2010–2011. The number of various health intervention programs has been increasing constantly worldwide since that time [53,54], which theoretically can promote responsibility for one’s own health. Similarly, increasing responsibility for one’s health influences technological interventions (among other health-related apps), which are more and more common recently [55]. However, the HLC is a personality trait and those in adults tend to be constant. We planned to conduct a follow-up study in 2020–2021, but the COVID-19 pandemic made it impossible to carry out the study. Further studies focused on the impact of HLC on physical activity and eating behaviors in various weight groups are necessary to confirm the relationships we observed. In addition, the relationships between types of locus of control and predisposition to undertake treatment for obesity, adherence to this treatment and obtained results should be assessed. Besides, the longitudinal assessment of the association between HLC and overweight or obesity development may be interesting for creating a strategy for prevention.

## 5. Conclusions

Women with obesity and with overweight more often have external health locus of control with both the impact of others and the impact of chance than women with normalweight. However, men with overweight and with obesity more often have only external health locus of control with the impact of others.

## Figures and Tables

**Figure 1 ijerph-19-15513-f001:**
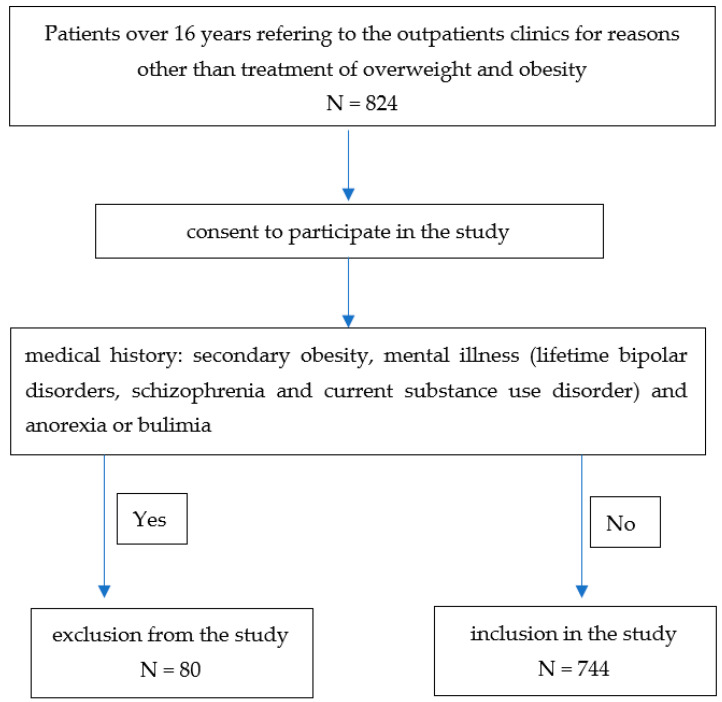
Study recruitment flow chart.

**Figure 2 ijerph-19-15513-f002:**
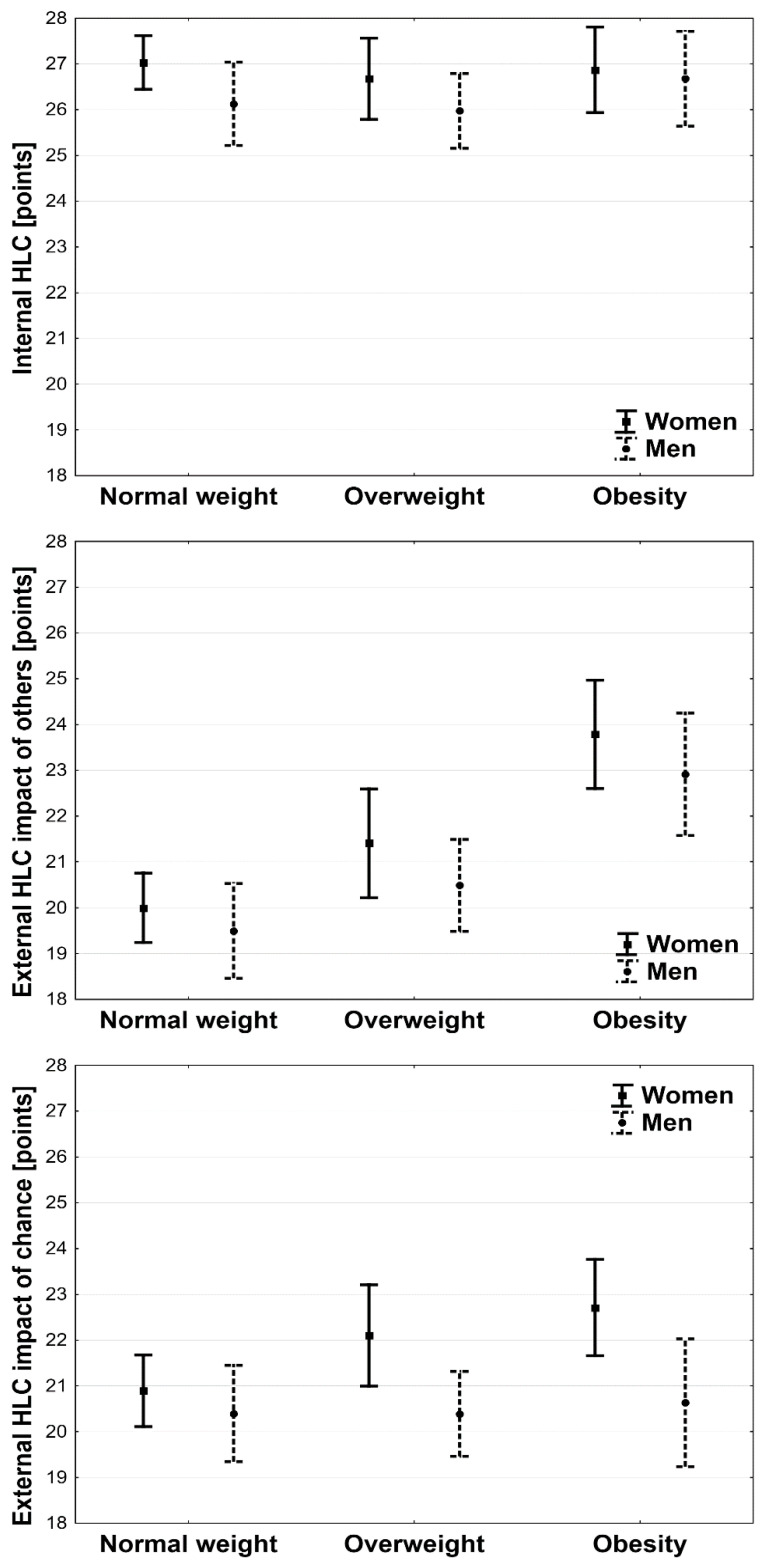
Health locus of control (HLC) and sex and body mass index (BMI).

**Figure 3 ijerph-19-15513-f003:**
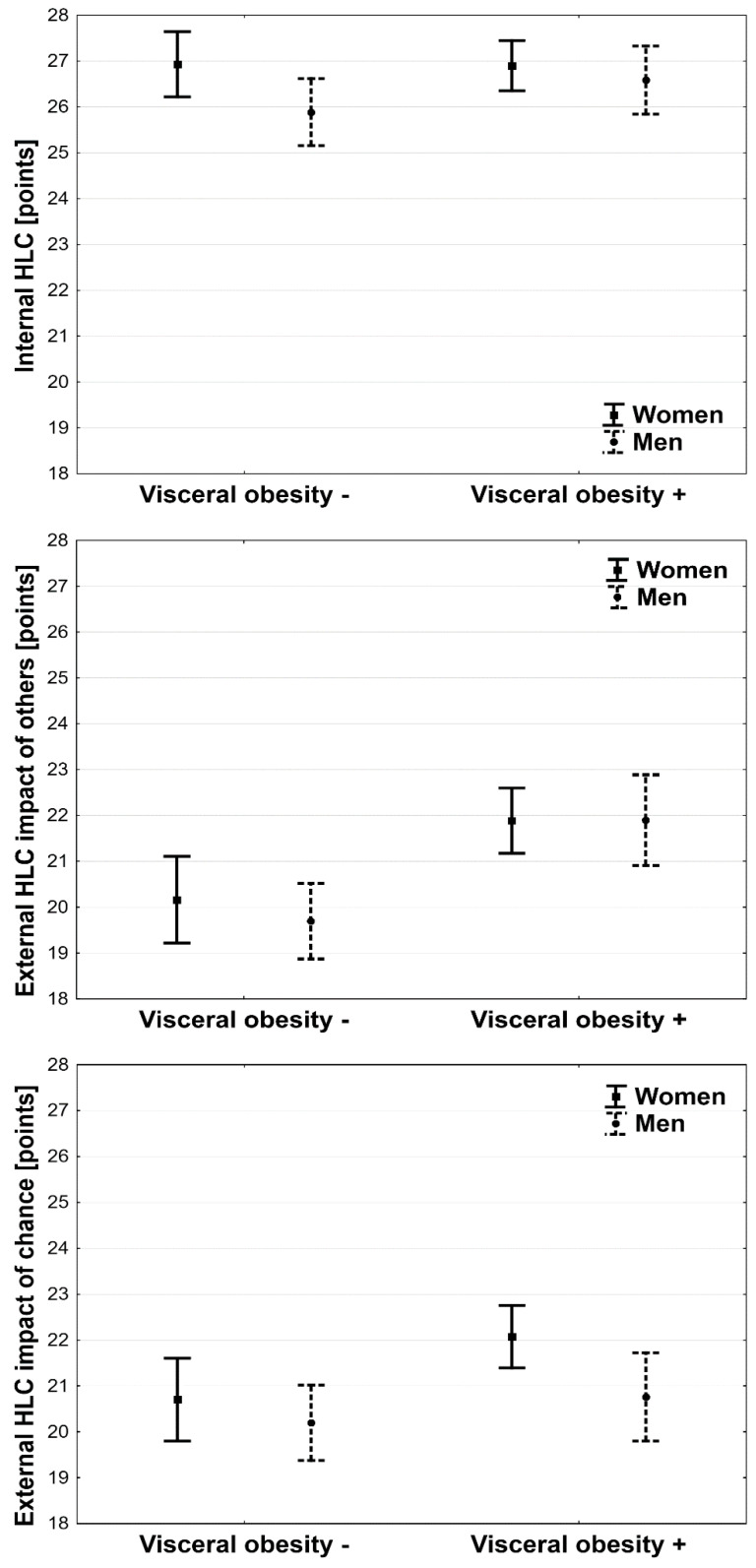
Health locus of control and sex and visceral obesity.

**Table 1 ijerph-19-15513-t001:** Basic characteristics of the study group according to nutritional status.

	Normal Weight N = 347	Overweight N = 221	Obesity N = 176	*p* for Trend
Age [years]	32 ± 10	36 ± 17	43 ± 13	<0.001
• Women	33 ± 11	40 ± 11	45 ± 12	<0.001
• Men	29 ± 9	33 ± 12	41 ± 14	<0.001
Higher education [N; %]	142; 41.2	49; 22.3	30; 17.3	<0.001
• Women	97; 41.1	22; 20.4	16; 15.1	<0.001
• Men	45; 41.3	27; 24.1	14; 20.9	<0.01
Marital status—living alone [N; %]	203; 60.2	89; 42.2	57; 34.1	<0.001
• Women	119; 52.2	33; 32.3	32; 31.4	<0.001
• Men	84; 77.1	56; 51.4	25; 38.5	<0.001
Alcohol consumption [N; %]	248; 71.5	160; 72.4	113; 64.2	0.13
• Women	151; 64.0	61; 56.5	58; 53.7	0.06
• Men	97; 87.4	99; 87.6	55; 80.9	0.27
Smoking [N; %]	157; 45.2	125; 56.6	107; 60.8	<0.001
• Women	98; 41.5	55; 50.9	62; 57.4	<0.01
• Men	59; 53.1	70; 62.0	45; 66.2	0.07
Active in sport [N; %]	114; 32.8	64; 29.0	47; 26.7	0.13
• Women	67; 28.4	22; 20.4	25; 23.1	0.20
• Men	47; 42.3	42; 37.2	22; 32.4	0.22

**Table 2 ijerph-19-15513-t002:** Results of the univariable and multivariable ordinal logistic regression for overweight and obesity.

	Women					Men				
**Univariable**	**OR**	**−95% CI**	**+95% CI**	**z**	** *p* **	**OR**	**−95% CI**	**+95% CI**	**z**	** *p* **
Age [each 5 years]	1.42	1.31	1.54	8.39	**<0.001**	1.33	1.21	1.46	5.99	**<0.001**
Higher education	1.40	0.94	2.08	1.67	0.10	1.19	0.75	1.88	0.75	0.46
Marital status—living alone	0.91	0.64	1.31	−0.49	0.62	0.84	0.55	1.29	−0.79	0.43
Smoking	0.84	0.59	1.19	−0.99	0.32	1.32	0.86	2.03	1.28	0.20
Alcohol consumption	0.70	0.49	1.00	−1.96	**0.05**	0.71	0.38	1.33	−1.07	0.29
Active in sport	0.74	0.49	1.12	−1.40	0.16	0.74	0.47	1.14	−1.35	0.18
Internal HLC [high]	0.91	0.64	1.30	−0.50	0.62	1.11	0.72	1.72	0.47	0.64
External HLC with the impact of others [high]	1.80	1.26	2.56	3.25	**<0.01**	1.86	1.20	2.88	2.80	**<0.01**
External HLC with the impact of chance [high]	1.59	1.10	2.22	2.47	**<0.05**	1.01	0.65	1.57	0.05	0.96
**Multivariable**	**OR**	**−95% CI**	**+95% CI**	**z**	** *p* **	**OR**	**−95% CI**	**+95% CI**	**z**	** *P* **
Age [each 5 years]	1.40	1.29	1.52	8.02	**<0.001**	1.32	1.20	1.46	5.78	**<0.001**
External HLC with the impact of others [high]	1.53	1.06	2.22	2.26	**0.05**	1.61	1.02	2.53	2.05	**<0.05**

## Data Availability

The datasets used and/or analyzed during the current study are available from the corresponding author upon reasonable request.

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
