# Peer review of "Can Nutritional Status in Adults Be Influenced by Health Locus of Control?"

_ijerph, 2022, doi:10.3390/ijerph192315513_

Round 1
Reviewer 1 Report (Previous Reviewer 3)
The authors aimed to assess the relationship between health locus of control and nutritional status in adults. Their results have demonstrated that the percentage of subjects with internal HLC did not differ significantly between obese, overweight and normal-weight groups. The percentage of subjects with external HLC with the impact of others was significantly (p<0.01) higher in both men and women with obesity than corresponding overweight and normal-weight groups. There are some critical issues should be addressed in this manuscript as follows.
1. First of all, the present manuscript is lack of novelty in terms of relationship between health locus of control and nutritional status in adults.
2. How to define the internal HLC and external HLC?
3. In Table 1, sampling is not objective enough, resulting in the biased explanation.
4. There is no clinical parameters to verify their findings.
5. Some of the sentences are difficult to understand and the grammar errors in the manuscript require further corrections.
Author Response
- First of all, the present manuscript is lack of novelty in terms of relationship between health locus of control and nutritional status in adults.
Ad. 1. In the introduction section was added a description of novelty: ‘So far, no studies have been conducted to assess the HLC in which categories of nutritional status are assessed on the basis of actual measurements of body weight and height in people of different ages.’
In addition in the discussion section, the following fragment was added: ‘To the best of our knowledge, this is the first study that assessed group not only on young subjects but also on the middle-aged and the elderly. Moreover, the assessment of BMI is based on measurements, and not on self-reported weight and height. Furthermore, it is the first study that analyzed the association between visceral obesity and HLC.’
- How to define the internal HLC and external HLC?
Ad 2. Definitions of internal HLC, as well as external HLC, were provided in the introduction section.
These definitions were improved to make the text more comprehensive: ‘External HLC is an assumption that the obtained health result depends on external factors, for example, physician; other person or chance. In other words, health-related events are perceived as not completely dependent upon personal actions. In contrast, internal HLC is the belief that health depends on the individual. Such a person believes in a causal relationship between actions taken and certain health-related events.’
- In Table 1, sampling is not objective enough, resulting in the biased explanation.
The capture and table presentation were changed. We agree that our study enrolled subjects utilizing medical services, which limits the generalization of the obtained results for the general population. We couldn’t avoid such a bias. Such a statement was added to the limitation section.
- There is no clinical parameters to verify their findings.
The aim of the study was to assess the relationship between health locus of control and nutritional status in adults. We assessed the relationship between health locus of control and body mass index as well as waist circumference. We did not assess the complications of obesity because the location of health control may only indirectly affect their occurrence. This was added to the limitations of the study.
- Some of the sentences are difficult to understand and the grammar errors in the manuscript require further corrections.
Ad 5. The manuscript was corrected once more by a native English speaker.
Reviewer 2 Report (New Reviewer)
The authors aimed to assess the relationship between health locus of control and adult nutritional status. The study included 744 adults (452 women, 292 men, 2.8% underweight, 43.8% 19 normal weight, 29.7% overweight, 23.7% with obesity). In addition to anthropometric measurement and socio-demographic data, health locus of control, using the Multidimensional Health Locus of Control Scale (MHLC) was assessed. This study suffers several limitations, including the adopted methodology, especially the experimental groups, the status of the study subjects, and the conclusion. However, a few questions on the work can be raised.
Among the 744 study subjects enrolled, the authors have taken the subjects aged 16 years or above as a single group. It would be better to have further sub-classifications to give a strong conclusion.
It is recommended to have Hierarchical regression to predict nutritional status (Mini-Nutritional Assessment (MNA) scores) in the study subjects.
Since this study focus on overweight and obesity, What about the selection criteria of diabetes among the study subjects?
The authors look into the grammatical errors in the manuscript.
Author Response
- Among the 744 study subjects enrolled, the authors have taken the subjects aged 16 years or above as a single group. It would be better to have further sub-classifications to give a strong conclusion.
Ad 1. Correlation analysis taking into account age was performed.
- It is recommended to have Hierarchical regression to predict nutritional status (Mini-Nutritional Assessment (MNA) scores) in the study subjects.
Ad 2. Please forgive us, but such an analysis is not possible, we did not collect the data contained in the MNA scale because it is a screening tool used to identify malnutrition or the risk of malnutrition in older adults.
- Since this study focus on overweight and obesity, What about the selection criteria of diabetes among the study subjects?
Ad 3. Because diabetes is a complication of obesity, the location of health control may only indirectly affect its occurrence, therefore we did not analyze the occurrence of this and other obesity complications in the study group. This was added to the limitations of the study.
- The authors look into the grammatical errors in the manuscript.
Ad 4. The manuscript was corrected once more by a native English speaker.
Round 2
Reviewer 1 Report (Previous Reviewer 3)
The authors have addressed most of my concerns.
Reviewer 2 Report (New Reviewer)
The authors response looks sound and made significant changes in the manuscript.
This manuscript is a resubmission of an earlier submission. The following is a list of the peer review reports and author responses from that submission.
Round 1
Reviewer 1 Report
The work is well designed and has a large sample size. The statistical procedures are correct and interesting results are obtained. However, it has certain formal defects that should be corrected.
Table 1 should be removed from the material and methods section and placed in the results section. Table 2 (whose caption is Health locus of control and visceral obesity) should be included in subsection 3.2 whose title is exactly the same. Moreover, table 2 and figure 2 contain redundant information.
In figure 1 the scale of the Y-axis should be the same in all three graphs. Now the first one has one decimal place, while the second and third ones have no decimal places.
In the discussion section, the authors start by saying that this paper analyses the association between health locus of control and body mass index, but this is not exactly the case, as they include visceral obesity as well. Therefore, I suggest that they write that they analyse the association between health locus of control and nutritional status.
The authors comment that there are participants aged 16 years and older. Please note that the World Health Organisation cut-off points you refer to (where a BMI of 25 and a BMI of 30 mark overweight and obesity respectively) cannot be applied to subjects aged 16 and 17 years . In this case, you should apply the reference values of the growth charts between 5 and 19 years published in 2007 by the WHO or the cut-off points published by Cole et. al. in 2000 in the British Medical Journal and recommended by the International Obesity Task Force. In any case, as there are probably not many individuals, to avoid repeating the categorisation of nutritional status, please comment on this in the limitations section.
Please remove the term non-Caucasian race and replace it with populations of non-European origin (or Asian and Afro-descendants etc).
Another issue concerns the use of the word gender to refer to men and women. I think you have made the classification on the basis of biological sex and not on the basis of the identity with which each person is considered. I think it would be more appropriate to change the word gender to sex. Finally, replace "patients" for participants.
Author Response
- The work is well designed and has a large sample size. The statistical procedures are correct and interesting results are obtained. However, it has certain formal defects that should be corrected. Table 1 should be removed from the material and methods section and placed in the results section. Table 2 (whose caption is Health locus of control and visceral obesity) should be included in subsection 3.2 whose title is exactly the same. Moreover, table 2 and figure 2 contain redundant information.
Ad 1: Table 1 was moved to the results section. We decided two leave both presentations of the data (table and figure). In fact table contain the exact mean values with SD, while the figure present mean vales with 95%CI.
- In figure 1 the scale of the Y-axis should be the same in all three graphs. Now the first one has one decimal place, while the second and third ones have no decimal places.
Ad 2: Done
- In the discussion section, the authors start by saying that this paper analyses the association between health locus of control and body mass index, but this is not exactly the case, as they include visceral obesity as well. Therefore, I suggest that they write that they analyse the association between health locus of control and nutritional status.
Ad 3: It was corrected throughout the manuscript.
- The authors comment that there are participants aged 16 years and older. Please note that the World Health Organisation cut-off points you refer to (where a BMI of 25 and a BMI of 30 mark overweight and obesity respectively) cannot be applied to subjects aged 16 and 17 years . In this case, you should apply the reference values of the growth charts between 5 and 19 years published in 2007 by the WHO or the cut-off points published by Cole et. al. in 2000 in the British Medical Journal and recommended by the International Obesity Task Force. In any case, as there are probably not many individuals, to avoid repeating the categorisation of nutritional status, please comment on this in the limitations section.
Ad 4: There were only 3 participants in age of 17 years. We have used the growth percentile charts for Polish population, but it does not change the nutritional status classification and in consequence the results.
- Please remove the term non-Caucasian race and replace it with populations of non-European origin (or Asian and Afro-descendants etc).
Ad 5: Was corrected.
- Another issue concerns the use of the word gender to refer to men and women. I think you have made the classification on the basis of biological sex and not on the basis of the identity with which each person is considered. I think it would be more appropriate to change the word gender to sex.
Ad 6: Was corrected throughout the manuscript.
- Finally, replace "patients" for participants.
Ad 7: The word “patients” was replaced for “participants” in the manuscript, when referring to the subjects included in the study.
Reviewer 2 Report
This manuscript compared HLC among people with or without normal weight and evaluate the associations of HLC with overall and central obesity in the population. My major concern is the strong confounding effect of age, which was not balanced across subgroups. My other suggestions include:
1) A flow chart for subject recruitment is needed to make it clear how the subjects were selcected.
2) More details on MHLC scale should be described in the manuscript. For example, what is impact of chance? what is impact of others? It would be better to present the items of the scale and the average score of the items in the study subjects. The validation of the scale in the study subjects is also needed to be presented.
3) Adjusted OR and 95%CI of HLC with obesity should be estimated.
4) The titles of the tables and figures in the manuscript would be improved to make it clear what the tables and figures talked about.
Author Response
- This manuscript compared HLC among people with or without normal weight and evaluate the associations of HLC with overall and central obesity in the population. My major concern is the strong confounding effect of age, which was not balanced across subgroups.
Ad 1: We add ordinal logistic regression analyses and multivariable model including age.
- A flow chart for subject recruitment is needed to make it clear how the subjects were selected.
Ad 2. Was added.
- More details on MHLC scale should be described in the manuscript. For example, what is impact of chance? what is impact of others?
Ad 3. These data are described in the introduction section.
- It would be better to present the items of the scale and the average score of the items in the study subjects.
Ad 4: The sub-scales of the MHLC questionnaire have been calculated with the computer software which provided the converted item-based results. So we have only values of three sub-scales presented in the paper and we are unable to obtain the subject's item values.
- The validation of the scale in the study subjects is also needed to be presented.
Ad 5.: Validation for Polish version was previously performed by JuszczyÅ„ski. The following fragment was added to the material and methods section “The internal consistency of MHCL established on the basis of Cronbach's alpha is 0.74 for internal HLC, 0.69 for external HLC with impact of chance and 0.54 for external HLC with impact of others. Reliability (r) assessed by the test-retest method (after six weeks) for internal HLC is 0.72; for external HLC with impact of chance 0.60 and for external HLC with impact of others 0.64 [27].”
- Adjusted OR and 95%CI of HLC with obesity should be estimated.
Ad 6: We add ordinal logistic regression analyses and multivariable model.
- The titles of the tables and figures in the manuscript would be improved to make it clear what the tables and figures talked about.
Ad 7: Done
Reviewer 3 Report
1. many obesity-related confounding factors, such as SES, mood, smoking, metabolic syndrome (except obesity), are not reported and not adjusted in the models. 2. HLC seems to be related to education, which should be adjusted too.Author Response
- many obesity-related confounding factors, such as SES, mood, smoking, metabolic syndrome (except obesity), are not reported and not adjusted in the models.
Ad 1: Data on SES, mood and metabolic syndrome were unfortunately not collected in the study group. We indicated this in the discussion/limitation section.
- HLC seems to be related to education, which should be adjusted too.
Ad 2: We add ordinal logistic regression analyses and multivariable model.
Round 2
Reviewer 1 Report
The authors have incorporated all the recommendations made to them and the work has improved significantly. I congratulate them and wish them success in their future work.
Reviewer 3 Report
The authors have improved the defects in the manuscript and it is accepted for publishing now.